# Identifying individuals at high risk for dementia in primary care: Development and validation of the DemRisk risk prediction model using routinely collected patient data

David Reeves [1,2☯*], Catharine Morgan [1☯], Daniel Stamate [1,3], Elizabeth Ford [4], Darren M. Ashcroft [1,5,6], Evangelos Kontopantelis [1,7], Harm Van Marwijk [4], Brian McMillan [1]

1 Division of Population Health, NIHR School for Primary Care Research, Centre for Primary Care, Health Services Research and Primary Care, University of Manchester, Manchester, United Kingdom, 2 Division of Population Health, Centre for Biostatistics, Health Services Research and Primary Care, University of Manchester, Manchester, United Kingdom, 3 Computing Department, Goldsmiths, University of London, London, United Kingdom, 4 Department of Primary Care and Public Health, Brighton and Sussex Medical School, Brighton, United Kingdom, 5 Division of Pharmacy and Optometry, NIHR Greater Manchester Patient Safety Research Collaboration, University of Manchester, Manchester, United Kingdom, 6 Centre for Pharmacoepidemiology and Drug Safety, School of Health Sciences, Faculty of Biology, Medicine and Health, University of Manchester, Manchester, United Kingdom, 7 Division of Informatics, Imaging and Data Sciences, University of Manchester, Manchester, United Kingdom

☯ These authors contributed equally to this work.
* david.reeves@manchester.ac.uk

## Abstract

### Introduction

Health policy in the UK and globally regarding dementia, emphasises prevention and risk reduction. These goals could be facilitated by automated assessment of dementia risk in primary care using routinely collected patient data. However, existing applicable tools are weak at identifying patients at high risk for dementia. We set out to develop improved risk prediction models deployable in primary care.

### Methods

Electronic health records (EHRs) for patients aged 60–89 from 393 English general practices were extracted from the Clinical Practice Research Datalink (CPRD) GOLD database. 235 and 158 practices respectively were randomly assigned to development and validation cohorts. Separate dementia risk models were developed for patients aged 60–79 (development cohort n = 616,366; validation cohort n = 419,126) and 80–89 (n = 175,131 and n = 118,717). The outcome was incident dementia within 5 years and more than 60 evidence-based risk factors were evaluated. Risk models were developed and validated using multivariable Cox regression.

**Data Availability Statement:** The data used in this study was obtained via the Clinical Practice

Research Datalink (CPRD). CPRD provides a research service which provides representative, longitudinal real-time anonymized patient electronic health records data from primary care and other health services across the UK. The licensing agreement between University of Manchester and CPRD, and the data governance of CPRD prevent the sharing or distribution of patient data to other individuals. Hence any requests for access to data from the study should be addressed to cprdenquiries@mhra.gov.uk. All researchers requiring access will require approval of their proposals from CPRD before data release.

**Funding:** This study was funded by Alzheimer's Research UK (ARUK: https://urldefense.com/v3/__ https://www.alzheimersresearchuk.org/__;!! PDiH4ENfjr2_Jw! EYvKRuQgEmAfSpu73NKZ9jFnv44lWSDi-DodiJn9z_

## Results

The age 60–79 development cohort included 10,841 incident cases of dementia (6.3 per 1,000 person-years) and the age 80–89 development cohort included 15,994 (40.2 per 1,000 person-years). Discrimination and calibration for the resulting age 60–79 model were good (Harrell's C 0.78 (95% CI: 0.78 to 0.79); Royston's D 1.74 (1.70 to 1.78); calibration slope 0.98 (0.96 to 1.01)), with 37% of patients in the top 1% of risk scores receiving a dementia diagnosis within 5 years. Fit statistics were lower for the age 80–89 model but dementia incidence was higher and 79% of those in the top 1% of risk scores subsequently developed dementia.

## Conclusion

Our models can identify individuals at higher risk of dementia using routinely collected information from their primary care record, and outperform an existing EHR-based tool. Discriminative ability was greatest for those aged 60–79, but the model for those aged 80–89 may also be clinical useful.

## Introduction

Aging populations in many parts of the world are set to greatly increase the numbers of older people living with dementia. In the United Kingdom (UK) dementia has an estimated prevalence of 7% amongst those aged over 65, with case numbers forecast to double between 2019 and 2040 to around 1.6 million [1], along with a tripling of the total costs of dementia care to society [1]. Furthermore, nearly 40% of all dementia cases go undiagnosed [2], a figure made worse by the COVID-19 pandemic [3]. International initiatives have been launched urging clinicians to be more pro-active in dementia diagnosis [4–6] and early diagnosis of dementia has been a key objective of the UK National Dementia Strategy since 2009 [7]. However, due to a lack of effective treatments for diagnosed dementia, more recent years have seen an increasing emphasis in both UK and global policy towards dementia prevention and risk reduction, and not just early detection [8].

In the UK, general practitioners (GPs)—and increasingly the wider primary care team—play a key role in the recognition and management of dementia in the community, and are financially incentivised to maintain dementia registers and provide recommended care [9]. The Lancet Commission in 2020 identified 12 modifiable risk factors which they estimated could account for up to 40% of dementia cases worldwide [10]. Many of these factors, including blood pressure, alcohol consumption, smoking, body-mass index, diabetes, depression and social isolation, are potentially amenable to interventions initiated in primary care, either directly or via referral to other services. To help GPs select patients who might benefit from early interventions, one option would be some form of dementia risk prediction tool that could readily identify individuals at higher risk for the disease, similar to other branches of medicine [11].

Many dementia risk models have been developed, though none are in general clinical usage [12, 13]. The current National Institute for Health and Care Excellence (NICE) guidance and quality statement on dementia are unclear about their prospective role [14]. Their ability to discriminate individuals at high risk has been variable and generally not strong [4, 12, 15]. Available models are typically multi-factorial, producing a risk score by combining across

psychological, sociodemographic, health, lifestyle, environmental and other factors. Most have been developed using data from longitudinal population-based surveys while a few utilise electronic health records (EHRs), sometimes combined with additional information collected directly from the patient [16, 17].

We are aware of just three risk models developed for a UK general population. Two of these —the UK Biobank Dementia Risk Score (UKBDRS) and the UKB Dementia Risk Prediction (UKB-DRP) model -were developed using the UK Biobank and both include some factors routinely recorded in a patient's primary care EHR but others that are not [18, 19]. A dementia risk prediction model computed solely from the EHR would arguably be preferable, that could be automated to quickly generate risk scores for purposes such as screening or stratifying the eligible practice population, as well as be part of regular health checks or general consultations. Models of this kind have been successfully developed in other disease areas, such as the QRISK tool for predicting risk of cardiovascular disease [20]. The third UK-focused model is of this type: the Dementia Risk Score (DRS), developed by Walters et al utilising the UK THIN primary care database [21]. The DRS consists of 14 predictive factors extracted from a patient's primary care EHR [22] and has good performance for patients aged 60–79 years (a C-statistic of 0.84), but not for those aged 80 or above (C-statistic of 0.56). However, the rate of true positives—the percentage of identified patients subsequently acquiring dementia, a key indicator of utility in practice—did not exceed 11% even at the highest levels of predicted risk.

The DRS investigated a relatively small number of potential predictors. In addition, model development and validation was based on EHR data pertaining to the youngest age (after turning 60 years) at which each individual met the criteria for inclusion, producing analysis cohorts very much dominated by patients in their early 60's. In actual primary care practice the population receiving risk assessment over any period of time is likely to include a much larger proportion of older people, especially when repeat assessments, such as at annual health checks, are taken into account.

Weak predictive performance may also be related to under-recording of incident dementia in the EHR, causing associations with risk factors to be under-estimated. As well as cases of undiagnosed dementia, EHRs can lack data on diagnoses made by another service provider, such as a hospital consultant or memory service specialist. A major UK source of additional diagnostic information is the NHS-Digital Hospital Episode Statistics (HES) dataset, which contains detailed data on all admissions to National Health Service (NHS) hospitals in England [23], including up to 20 existing clinical diagnoses. To the best of our knowledge, existing dementia risk models developed on UK primary case databases have not included linkage to external diagnostic sources.

Our study aims were to develop risk models with improved ability to identify patients at higher risk of future dementia from their UK primary care record, by investigating a wider set of potential predictive factors, constructing development and validation cohorts more representative of the target population, and by increasing the completeness of information on dementia diagnosis using linked secondary care data.

## Methods

### Study design and data source

We utilised the Clinical Practice Research Datalink (CPRD) GOLD primary care database to develop and validate models to predict risk of newly recorded dementia within 5 years. CPRD GOLD is a large anonymised EHR database, broadly representative of the UK population [24]. During the study period GOLD had more than 700 contributing general practices covering approximately 8% of the UK population [25]. Data is recorded using the Read code system in

regard to consultations, symptoms, diagnoses, investigations, biometrics, prescriptions, treatments and referrals [26]. Access to CPRD GOLD was obtained under licence from the UK Medicines and Healthcare products Regulatory Agency. CPRD GOLD consists of routinely collected data that has been pseudonymised for the purposes of research, for which informed consent was not required. The study was approved by the independent scientific advisory committee for Clinical Practice Research Datalink research (protocol No 18_163R).

GOLD was linked to two additional datasets: the NHS-Digital Hospital Episodes Statistics Admitted Patient Care (HES-APC) dataset; and the Office for National Statistics (ONS) index of multiple deprivation (IMD) 2010. For each patient, we identified the earliest recorded date of dementia diagnosis in either GOLD or HES-APC. Dementia in the HES was defined using a set of ICD-10 codes from the National Audit of Dementia hospital survey in 2016/17 (Table in S2 File). [26] Compared against a large mental health registry, HES dementia diagnoses have been found to have good sensitivity (78%) and specificity (92%) [27].

The IMD is a UK Government small-area (approximately 1500 people) composite deprivation score combining seven indices relating to income, employment, education, health, living environment, access to services and crime [28]. Dementia is more prevalent in areas of higher social deprivation [29]. IMD scores are available for practice locations and for patient residences. These are highly correlated but neither is routinely recorded in the EHR. However, the former can be implemented in a predictive model as a constant value for all patients within a given practice. As a practice-level characteristic, this factor may also help account for heterogeneity between sites [30].

We identified a total of 696 practices in GOLD with data of acceptable quality ("up to standard", as defined by the CPRD organisation) across our study period. Linked HES-APC and ONS data were available for 393 of these (56%), all based in England. Analysis was restricted to this group. Within quintiles of practice IMD scores we randomly assigned 60% of practices (total = 235) to a development cohort and the other 40% (158) to a separate validation cohort. In view of the large sample we opted for a fairly large validation cohort to ensure accurate estimates of model performance indices.

## Participants

We included registered patients aged between 60 and 89 (i.e. had not reached their 90th birthday) contributing data to GOLD between 1st January 2005 and 31st December 2017. Patients under age 60 at 1st January 2005 entered the study at a later date when they reached age 60 provided they met all other inclusion criteria. We excluded patients with a code for dementia recorded prior to study entry, with less than one year of continuous registration in the practice prior to study entry, or with less than one full year of consultation data prior to study entry. We also excluded patients with prior conditions associated with the development of dementia-like symptoms (Parkinson's disease, Huntington's disease, HIV infection, Creutzfeldt-Jakob disease, Pick's disease, Lewy body, dementia in other conditions, alcohol and drug-related dementia) and those with a cognitive impairment or memory loss code within the previous 5 years potentially indicative of prodromal or unrecorded dementia (Tables in S2 File).

Previous research has suggested a disjunction in the risk of dementia at around 80 years of age [17, 22]. In line with this we constructed two separate age cohorts for analysis, consisting of age-groups 60–79 and 80–89 at the index date, and conducted separate model development and validation for each age-group. Patients who crossed the age-threshold of 80 years within the analysis window of 2005 to 2017 were included in both age cohorts where eligible (Fig 1).

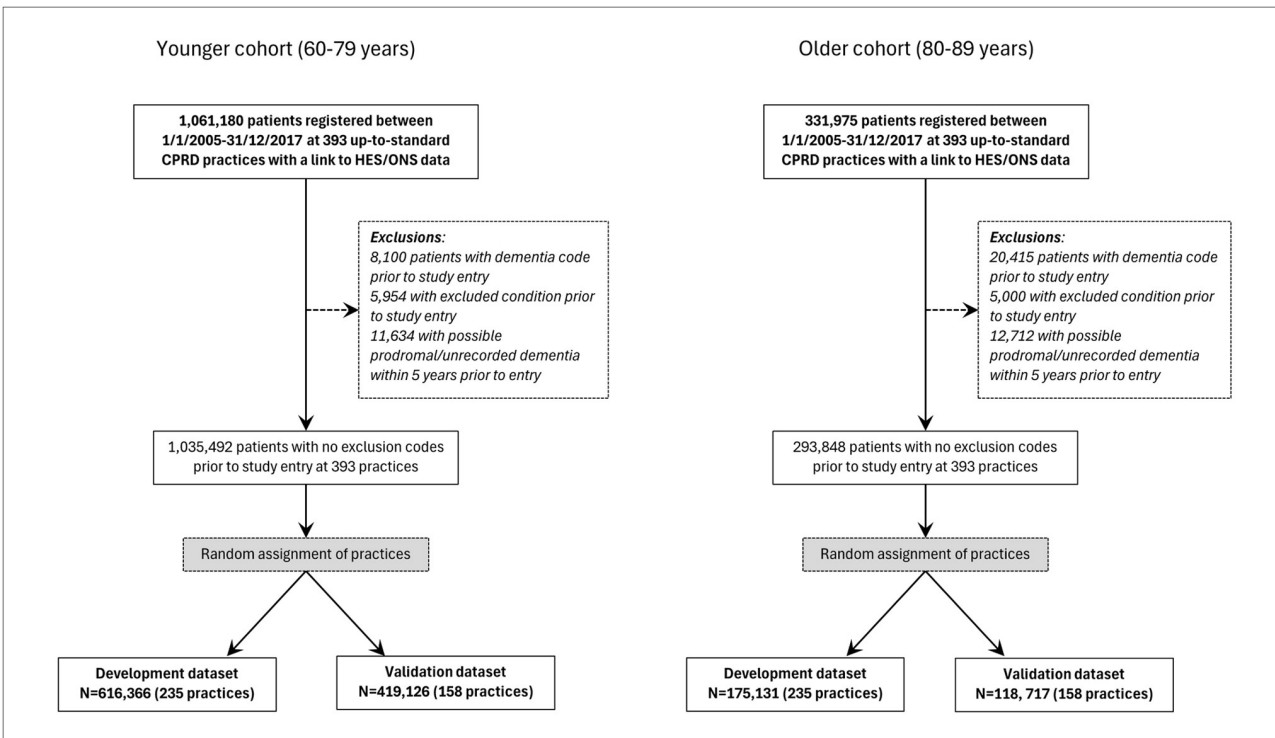

**Fig 1. Flowcharts of the construction of the development and validation cohorts for each age-group.**

## Follow-up period

Follow-up was the period between the date a patient became eligible for the study and the date they exited the cohort, defined as the earliest of diagnosis of dementia, death, transfer out, practice left CPRD or 31st December 2017. Follow-up was restricted to a maximum of 5 years and ceased at this point or when the patient exited the cohort, if earlier. This period was divided into 12-monthly "year-bands" and each individual assigned an "index date" as the starting date of a randomly selected year-band. This produced a cohort with an age distribution broadly in line with the patient population eligible for risk assessment across the time period. In particular, in the resulting 60–79 cohort patients aged 60 years constituted just 12% of the total, compared to 37% when using each individual's earliest date of eligibility (Fig A1 in S1 File).

## Outcome

The outcome was a new diagnosis of dementia within 5 years of a patient's index date, identified by a relevant Read code in the primary care record or ICD-10 code in the linked secondary-care HES-APC dataset. Relevant Read codes were identified in a consensus exercise by a panel of clinicians, academic GPs and pharmacologists at Manchester and included all types of dementia diagnoses but excluded diagnoses associated with Parkinson's disease, Huntington's disease, HIV, Creutzfeldt-Jakob disease, Pick's disease, Lewy body, alcohol and drug-related dementia. The definition also included a set of drug therapy codes prescribed exclusively for dementia symptoms (Tables in S2 File).

## Predictive factors

We developed a list of candidate predictive factors based on published systematic reviews of dementia risk factors and of dementia risk prediction models [12, 13, 31], selecting factors for which the evidence base supported a relationship to dementia risk and that were amenable to construction within GOLD (Table A1 in S1 File). We excluded factors that could be indicative of prodromal dementia or unrecorded dementia (e.g. memory loss; cognitive testing) or that are poorly or unreliably recorded in the CPRD, such as education, family history of dementia and ethnicity especially for older patients [32]. We included a few additional factors identified from other published studies. Where the evidence suggested that a medication class might potentially represent a risk factor in itself we included the class separately to the medical condition for which it was prescribed. We also included measures of polypharmacy (number of different prescribed medicines over previous 12 months) and of anticholinergic burden (by which each drug is assigned a score of 1, 2 or 3 depending upon degree of anticholinergic effect and the scores totalled) [33].

Most predictive factors were operationalised as Yes/No binary variables: either appearing in the EHR prior to the index date or not. These were mainly medical conditions (e.g. diabetes) and prescribed medications. Other factors took the form of categorical variables (e.g. practice IMD, smoking status), counts (e.g. polypharmacy count, number of A&E visits and number of home visits, in last 12 months) or continuous biometric measurements (BMI, systolic BP, diastolic BP, pulse pressure, serum cholesterol). Continuous and count factors were subjected to initial fractional polynomial analysis using a Cox model, to determine which transformation (with terms restricted to untransformed, square-root, log, squared or cubed) had the greatest predictive ability [34]. For most factors this was the square-root transformation. For age it was age and age-squared.

Recording of biometric factors can be sporadic and varies by practice. For BMI we used the most recent recorded value. Serum cholesterol and blood/pulse pressure measurements are more variable and in some cases were not measured each year or measured multiple times within a year. For these factors we took the mean value across the most recent year in which they were measured. These factors were coded as missing for patients with no recorded measurement.

For each patient, all predictive factors were constructed based on the EHR data prior to their index date.

## Analysis

### Adequacy of sample size

Our sample numbers were determined by the size of the GOLD dataset. We therefore followed the suggestion of Riley et al [35] and applied their Stata pmsampsize package to estimate the maximum number of variables we could evaluate in a model whilst keeping overfitting acceptably low (small optimism in the factor coefficient estimates and a minimally inflated $R^2$) and error of no more than 0.05 on the estimate of overall risk at 5 years [36]. For this purpose Royston D values of 2.03 and 0.86 reported by Walters et al [22] for their younger and older validation cohorts were converted into Cox-Snell $R^2$ estimates for our samples, and to account for the nesting of patients within practices we estimated sample inflation factors of 4.7 and 3.5 respectively (based on calculated intra-cluster correlation coefficients (ICCs) for incident dementia of 0.0014 and 0.0069). Using these values together with the incident and follow-up rates in our development samples, we calculated that our younger cohort comfortably allowed for estimation of models with more than 200 factors (minimum required sample n = 492,936,

including 8,253 events) and our older cohort 100 factors (minimum sample n = 135,202, including 12,278 events).

## Missing data

Data was missing for some individuals for a small number of factors, with the highest rates being for serum cholesterol (23%) and BMI (10.0%). We used the Fully Conditional Specification (FCS) method of multiple imputation to impute missing data for BMI, serum cholesterol, systolic and diastolic blood pressure as continuous factors, and for smoking as a categorical factor. All other factors had complete data. Imputation was done separately for the development and validation cohorts and included the full set of candidate variables plus the outcome at 5 years along with the cumulative hazard function [37, 38]. Ten multiple imputation datasets were generated and results were pooled using Rubin's rules [38]. Variable removal in the backwards stepwise procedure was based on the pooled model at each step [38].

## Statistical models

We applied Cox proportional hazards regression models to build our predictive models, using time to dementia diagnosis as the outcome. Using the development cohort, we first conducted univariable analysis of each predictive factor whilst controlling for age and gender, to reduce collinearity with these key factors. Age in particular is highly associated with increasing likelihood of both incident dementia and most other types of health events.

We next ran a multivariable Cox analysis using the full set of factors. The performance of this "full model" when applied to the development data was assessed on the basis of predictions made at the randomly selected year-band for each individual. Performance was assessed in terms of Harrell's C (measures the probability that of a randomly selected pair of patients, the patient with the shorter survival time has the higher predicted risk [39]) and Royston's D (a measure of the separation between the survival curves for cases with higher and lower predicted risk, where a higher D indicates greater separation [40]). We also calculated precision, specified as the percentage of cases in the top 1% and 5% of estimated risk scores that acquired dementia within 5 years. With imbalanced datasets, fit indices are dominated by large numbers of non-cases; precision (which is equivalent to the true positive rate) provides a straight-forward measure of positive identification [41].

The full model was next subjected to a process of model reduction using a curated backwards stepwise procedure. The aim was to identify a smaller subset of predictive factors that resulted in only minimal reduction in performance compared to the full model. The reduction was curated in that decisions about factors to be dropped or retained at each step were based not only on statistical considerations (p-values, hazard ratios and multicollinearity), but also on clinical and practical considerations to ensure that the reduced model made clinical sense and could be readily automated in primary care [38]. In all analyses variance terms (and associated 95% confidence intervals) for all regression coefficients and fit indices took account of the clustering of patients within practices.

Our datasets are multi-level, consisting of patients within GP practices, and also include a practice-level risk factor, IMD. However, our principal analyses used single-level Cox regression with robust estimates of variance to allow for clustering within practices. This was in view of the very high computing overheads and risk of non-convergence when fitting multi-level models. Therefore as a sensitivity we reran our final risk models using multi-level Cox regression to assess whether this altered predictive performance at all. All analysis was conducted in Stata version 17 between January 2019 and August 2024.

## Validation and calibration

The resulting reduced model was applied to the validation cohort to assess performance in an independent set of GP practices, again on the basis of Harrell's C, Royston's D and precision. The calibration slope was estimated by using a Cox model to regress the binary outcomes in the validation cohort on the linear (log) predictor from the model [42]. A slope close to 1.0 indicates good calibration. Linearity in the calibration was assessed by plotting the mean predicted risks (x-axis) within 5 years against the observed risk (y-axis) in the validation cohort within deciles of predicted risk, where observed risks were obtained using the Kaplan-Meier estimates evaluated at 5 years.

## Risk classification

To provide a clearer picture of how the final models might perform in clinical practice, we assessed their ability to correctly identify patients with subsequent incident dementia, at a range of thresholds for high risk from 3% up to 50%. For each threshold we calculated the sensitivity, specificity, number and rate of true positives (TP; equivalent to the positive predictive value) and number and rate of false negatives (FN). Being based on non-censored cases only, these metrics tend to be distorted. Therefore we also modelled TP and FN values adjusted for censoring by weighting each case by the estimated probability of dementia. In addition, using the year-band level data, we modelled the yearly TP and FN rates expected were patients to receive an annual risk assessment.

## Comparison with other risk models

To compare the discriminant ability of our final model to that of the DRS developed by Walters et al [22], we implemented the DRS in our GOLD dataset. To make a like-for-like comparison we re-calibrated the DRS, using that model's factor-set whilst keeping key analysis specifications as per our models (i.e. our definition of dementia including HES-APC dementia diagnoses; model development and validation using random year-bands). Differences in the coding schemes underlying GOLD and THIN resulted in some minor differences in how some factors were specified.

We also compared the performance of our models to that of a model consisting of just age and age-squared as risk factors. As the single strongest risk factor for dementia, some studies adopt an age-only model as a "baseline" against which to compare more complex models. A recent evaluation of the external validity of several prediction models using the UK Biobank found the DRS and an age-only model to differ very little in discriminative ability [19], while a Finnish study that externally validated 4 prediction models, including the DRS, found that none performed much better than age alone [43].

# Results

## Cohort aged 60–79 years

**Development cohort 60–79 years.** There were 616,366 individuals in the 60–79 development cohort, with a mean age at Index of 67.9 years (SD 6.4) and 48.8% male (Table 1). The median length of follow-up was 2.63 years (inter-quartile range 0.96 to 5.0 years) and there were 10,841 incident diagnoses of dementia within 5 years over a total of 1,717,179 person years at risk (crude incidence rate of 6.31 per 1,000 person-years). 24% were incident cases of Alzheimer's disease, 18% vascular dementia, and 58% mixed, unspecified or other. Just over 30% of all incident cases were identified from the linked HES-APC dataset.

**Table 1. Characteristics of development and validation cohorts of patients aged 60–79 at their index dates.**

|  | Development cohort | | Validation cohort | |
|---|---|---|---|---|
| Number of GP practices | 235 | | 158 | |
| Total number of patients | 616,366 | | 419,126 | |
| Median length of FU (5 years max) | 2.63 | | 2.67 | |
| Interquartile range of FU | (0.96, 5) | | (0.99, 5) | |
| Number of incident dementia diagnoses (within 5 year FU) | 10,841 | | 7,425 | |
| Total person-years at risk | 1,717,179 | | 1,176,506 | |
|  | N (%) | Mean (SD) | N(%) | Mean (SD) |
| **Demographic and lifestyle factors** | | | | |
| Sex | | | | |
| Male | 300,522 (48.8) | - | 203,324 (48.5) | - |
| Female | 315,844 (51.2) | - | 215,802 (51.5) | - |
| Age | - | 67.9 (6.4) | - | 67.9 (6.4) |
| Practice IMD (quintiles) | | | | |
| 1 (lowest deprivation) | 95,199 (15.5) | - | 75,736 (18.1) | - |
| 2 | 124,561 (20.2) | - | 84,495 (20.2) | - |
| 3 | 129,483 (21.0) | - | 78,116 (18.6) | - |
| 4 | 129,839 (21.1) | - | 88,118 (21.0) | - |
| 5 (highest deprivation) | 137,284 (22.3) | - | 92,661 (22.1) | - |
| Smoking status | | | | |
| Never smoked | 218,519 (35.5) | - | 154,985 (37.0) | - |
| Ex-smoker | 251,434 (40.8) | - | 167.407 (39.9) | - |
| Current Smoker | 124,759 (20.2) | - | 83,223 (19.9) | - |
| Not recorded | 21,654 (3.5) | - | 13,511 (3.2) | - |
| Heavy drinking/alcohol problem ever | 92,542 (15.0) | - | 59,680 (14.2) | - |
| **Medical conditions (ever recorded)** | | | | |
| Anxiety | 122,272 (19.8) | - | 82,780 (19.8) | - |
| Depression | 122,347 (19.9) | - | 84,037 (20.1) | - |
| Epilepsy | 9,441 (1.5) | - | 6,476 (1.6) | - |
| Angina | 44,609 (7.2) | - | 27,690 (6.6) | - |
| Atrial fibrillation | 26,602 (4.3) | - | 18,017 (4.3) | - |
| Coronary bypass surgery | 12,027 (2.0) | - | 8,186 (2.0) | - |
| Cardiomyopathy | 2,204 (0.4) | - | 1,556 (0.4) | - |
| Coronary Heart Disease | 115,256 (18.7) | - | 75,151 (17.9) | - |
| Heart failure | 16,707 (2.7) | - | 11,458 (2.7) | - |
| Myocardial infarction | 29,078 (4.7) | - | 19,953 (4.8) | - |
| Stroke/TIA | 32,597 (5.3) | - | 21,737 (5.2) | - |
| Cancer (any) | 92,143 (15.0) | - | 63,319 (15.1) | - |
| Diabetes | 73,547 (11.9) | - | 50,126 (12.9) | - |
| Irritable bowel disease | 8,964 (1.5) | - | 6,251 (1.5) | - |
| Renal disease/failure | 45,546 (7.4) | - | 31,650 (7.6) | - |
| Fracture (any) | 129,207 (21.0) | - | 87,026 (20.8) | - |
| Gait disturbance | 9,107 (1.5) | - | 6,164 (1.5) | - |
| Mild traumatic head injury (incl. concussion) | 22,707 (3.7) | - | 15,127 (3.6) | - |
| Major traumatic head injury | 29,247 (4.8) | - | 19,341 (4.6) | - |
| Hearing impairment | 87,396 (14.2) | - | 58,784 (14.0) | - |
| Sight impairment | 8,713 (1.4) | - | 5,930 (1.4) | - |
| Sleep disturbance | 71,516 (11.6) | - | 49,164 (11.7) | - |

*(Continued)*

**Table 1.** (Continued)

| | | | | |
|---|---|---|---|---|
| Hyperlipidaemia | 129,489 (21.0) | - | 87,292 (20.8) | - |
| Hypertension | 255,608 (41.5) | - | 175,746 (41.9) | - |
| Hypotension | 8,232 (1.3) | - | 5,592 (1.3) | - |
| **Prescribed medications (ever prescribed)** | | | | |
| Anti-depressants | | | | |
| Selective serotonin reuptake inhibitor | 107,3025 (17.4) | - | 72,836 (17.4) | - |
| Tricyclic antidepressant | 134,356 (21.8) | - | 89,094 (21.3) | - |
| Other antidepressant | 27,091 (4.4) | - | 19,316 (4.6) | - |
| Antipsychotics | 106,248 (17.2) | - | 72,458 (17.3) | - |
| Benzodiazepines | 129,725 (21.1) | - | 87,256 (20.8) | - |
| Hypnotics | 157,032 (25.5) | - | 106,418 (25.4) | - |
| Mood stabilisers | 19,774 (3.2) | - | 13,778 (3.3) | - |
| Z-drugs | 54,398 (8.8) | - | 37,799 (9.0) | - |
| Aspirin | 167,011 (27.1) | - | 111,152 (26.5) | - |
| H2 receptor antagonists | 99,012 (16.1) | - | 64,974 (15.5) | - |
| Proton pump primers | 232,056 (37.7) | - | 157,072 (37.5) | - |
| NSAIDs (excluding aspirin) | 394,831 (64.1) | - | 266,311 (63.5) | - |
| Opioids | 88,163 (14.3) | - | 59,127 (14.1) | - |
| Statins | 226,163 (36.7) | - | 151,886 (36.2) | - |
| Lipid lowering medication (including statins) | 228,953 (37.2) | - | 153,948 (36.7) | - |
| Anti-hypertensives | 328,036 (53.2) | - | 224,026 (53.5) | - |
| Any anticholinergic | 336,566 (54.6) | - | 227,430 (54.3) | - |
| Anticholinergic burden score over last year | | 0.62 (1.42) | | 0.62 (1.42) |
| Polypharmacy count over last year | | 3.43 (4.26) | | 3.41 (4.26) |
| **Service interactions** | | | | |
| Ever received social care | 1,627 (0.3) | - | 1,142 (0.3) | - |
| Number of visits to A&E in last year | - | 0.11 (0.44) | - | 0.11 (0.44) |
| Number of GP consultations involving third party (e.g. relative/friend) in last year | - | 0.03 (0.31) | - | 0.03 (0.29) |
| Number of GP home visits in last year | - | 0.11 (1.07) | - | 0.11 (1.20) |
| Number of missed GP appointments in last year | - | 0.12 (0.44) | - | 0.13 (0.47) |
| **Biometric measures** | | | | |
| BMI ever recorded | 556,771 (90.3) | - | 375,589 (89.6) | - |
| Most recently recorded BMI | - | 27.5 (5.2) | - | 27.5 (5.3) |
| Total serum cholesterol ever recorded | 473,802 (76.9) | - | 320,247 (76.4) | - |
| Most recent total serum cholesterol (mmol/L)[a] | - | 5.2 (1.1) | - | 5.2 (1.1) |
| Blood pressure ever recorded | 592,556 (96.1) | | 405,540 (96.8) | - |
| Most recent systolic BP (mmHg)[a] | - | 137.4 (15.5) | | 137.7 (15.5) |
| Most recent diastolic BP (mmHg)[a] | - | 79.2 (8.7) | | 79.2 (8.8) |
| Most recent pulse pressure(mmHg)[a] | - | 58.2 (12.9) | - | 58.5 (13.0) |

IMD, Index of Multiple Deprivation

[a]Mean within the most recent year-band with recorded values

**Univariable associations between risk factors and incident dementia.** Increasing age and female sex demonstrated relationships with incident dementia (Table 2). After controlling for these, most other factors were associated with increased risk of dementia. The strongest associations were with a history of stroke/TIA, epilepsy, gait problems, use of antidepressants and mood stabilisers, anticholinergic burden, previous or current receipt of social care, and

**Table 2. Summary of univariable analyses of predictive factors, age 60–70 development cohort.**

| | 60–79 development cohort | |
|---|---|---|
| | **Hazard ratio** | **95% CI** |
| **Demographic and lifestyle factors** | | |
| Sex (Female) | 1.194[a] | 1.149 to 1.240 |
| Age—60 | 1.296 | 1.273 to 1.320 |
| (Age—60)^2 | 0.995 | 0.994 to 0.996 |
| Subsequent factors are controlled for age and sex | | |
| Calendar year (-2005) | 0.971 | 0.963 to 0.980 |
| Practice IMD quintile | | |
| 1 Lowest deprivation | Ref | |
| 2 | 1.000 | 0.895 to 1.120 |
| 3 | 1.003 | 0.898 to 1.119 |
| 4 | 1.077 | 0.957 to 1.211 |
| 5 highest deprivation | 1.290 | 1.142 to 1.456 |
| Smoking status, most recent[a] | | |
| Non-smoker | Ref | |
| Ex-smoker | 1.084 | 1.036 to 1.133 |
| Current smoker | 1.270 | 1.189 to 1.350 |
| Heavy drinking/alcohol problem ever | 1.178 | 1.103 to 1.258 |
| **Medical conditions (ever recorded)** | | |
| Anxiety | 1.350 | 1.294 to 1.408 |
| Depression | 1.618 | 1.544 to 1.696 |
| Epilepsy | 2.344 | 2.088 to 2.633 |
| Angina | 1.227 | 1.160 to 1.299 |
| Atrial fibrillation | 1.285 | 1.203 to 1.372 |
| Coronary bypass surgery | 1.122 | 0.999 to 1.261 |
| Cardiomyopathy | 1.090 | 0.736 to 1.614 |
| Coronary Heart Disease | 1.154 | 1.101 to 1.210 |
| Heart failure | 1.361 | 1.256 to 1.475 |
| Myocardial infarction | 1.127 | 1.050 to 1.210 |
| Stroke/TIA | 2.048 | 1.944 to 2.158 |
| Cancer (any) | 0.982 | 0.930 to 1.036 |
| Diabetes | 1.410 | 1.343 to 1.480 |
| Irritable bowel disease | 1.021 | 0.889 to 1.173 |
| Renal disease/failure | 1.165 | 1.098 to 1.234 |
| Fracture (any) | 1.184 | 1.130 to 1.240 |
| Gait problems | 1.965 | 1.781 to 2.167 |
| Mild traumatic head injury (incl. concussion) | 1.649 | 1.523 to 1.786 |
| Major traumatic head injury | 1.594 | 1.482 to 1.716 |
| Hearing impairment | 1.084 | 1.034 to 1.136 |
| Sight impairment | 1.411 | 1.267 to 1.570 |
| Sleep disturbance | 1.227 | 1.161 to 1.297 |
| Hyperlipidaemia | 1.070 | 1.027 to 1.116 |
| Hypertension | 1.038 | 0.996 to 1.082 |
| Hypotension | 1.563 | 1.376 to 1.776 |
| **Prescribed medications (ever prescribed)** | | |
| Anti-depressants | | |
| Selective serotonin reuptake inhibitor | 1.955 | 1.865 to 2.050 |

(*Continued*)

**Table 2.** (Continued)

|  | 60–79 development cohort | |
| --- | --- | --- |
|  | **Hazard ratio** | **95% CI** |
| Tricyclic antidepressant | 1.303 | 1.245 to 1.365 |
| Other antidepressant | 2.120 | 1.963 to 2.289 |
| Antipsychotic | 1.351 | 1.285 to 1.421 |
| Benzodiazepines | 1.224 | 1.167 to 1.283 |
| Hypnotics | 1.281 | 1.226 to 1.339 |
| Mood stabiliser | 1.984 | 1.837 to 2.142 |
| Z-drugs | 1.354 | 1.257 to 1.460 |
| Aspirin | 1.354 | 1.304 to 1.406 |
| H2 receptor antagonists | 1.060 | 1.009 to 1.114 |
| Proton pump primers | 1.070 | 1.027 to 1.115 |
| NSAIDs (excluding aspirin) | 0.927 | 0.887 to 0.970 |
| Opioids | 1.221 | 1.152 to 1.294 |
| Statins | 1.233 | 1.190 to 1.277 |
| Lipid lowering medication (including statins) | 1.233 | 1.190 to 1.277 |
| Antihypertensive | 1.199 | 1.150 to 1.250 |
| Anticholinergics (ever) | 1.282 | 1.229 to 1.338 |
| Anticholinergics burden over last year (square root) | 1.305 | 1.272 to 1.340 |
| Polypharmacy count over last year | 1.050 | 1.045 to 1.054 |
| **Service interactions** | | |
| Ever received social care | 3.804 | 3.138 to 4.611 |
| Number of A&E visits in last year (square root) | 1.712 | 1.620 to 1.810 |
| Number of GP consultations involving third party in last year | 2.097 | 1.863 to 2.360 |
| Number of GP home visits in last year | 1.503 | 1.410 to 1.603 |
| Number of missed GP appointments in last year | 1.874 | 1.784 to 1.969 |
| **Biometric measures** | | |
| Most recent BMI (square root)[a] | 0.665 | 0.631 to 0.698 |
| Most recent mean Serum Cholesterol[ab] | 0.960 | 0.941 to 0.980 |
| Most recent mean Systolic BP/20[abc] | 0.876 | 0.850 to 0.902 |
| Most recent mean Diastolic BP/20[abc] | 0.823 | 0.782 to 0.863 |
| Most recent mean Pulse Pressure/20[abc] | 0.905 | 0.874 to 0.937 |

IMD, Index of Multiple Deprivation

[a]Pooled across 10 datasets with missing data values imputed

[b]Mean within the most recent year-band with recorded values

[c]Rescaled by dividing by 20

frequent presence of a third party (e.g. a relative) at consultations. There was also a strong but negative association with BMI. We explored the inclusion of an interaction between blood pressure and use of anti-hypertensive medicines but found that this did not increase the overall degree of association. The assumption of proportional hazards was examined using scaled Schoenfeld residuals and log-log survival plots [44]. Correlations between the residuals and time to failure were all very small (rho< = 0.04). Likewise log-log survival plots did not suggest any notable deviations from proportional hazards.

**Table 3. Age 60–79 cohort, fit indices (and 95% confidence intervals) for the full and reduced models.**

| Model | Dataset | Harrell's C | Royston D | Top 1% precision[a] | Top 5% precision[a] | Calibration slope |
|---|---|---|---|---|---|---|
| Full multivariable model | Development cohort | 0.791 (0.787 to 0.796) | 1.80 (1.75 to 1.85) | 38.3% | 24.3% | NA |
| Reduced model | Development cohort | 0.787 (0.782 to 0.792) | 1.80 (1.76 to 1.84) | 40.0% | 25.0% | NA |
| Reduced model | Validation cohort | 0.781 (0.776 to 0.786) | 1.74 (1.70 to 1.78) | 36.7% | 22.8% | 0.981 (0.956 to 1.006) |

[a]Proportion of patients in the top 1%/5% of risk scores with incident dementia within 5 years

### Multivariable analysis

A Cox regression model using the full set of factors had a Harrell's C-statistic of 0.791 (95% CI: 0.787 to 0.796), Royston D of 1.80 (1.75 to 1.85) and top 1% (5%) precision of 38.3% (24.3%) (Table 3).

Following stepwise removal, a solution was obtained that reduced the factors in the model by around two-thirds without noticeably lowering performance within the development cohort (Table 3: Harrell's C = 0.787 (0.782 to 0.792); Royston D = 1.80 (1.76 to 1.84); 1% (5%) precision = 40% (25.0%)). Patient factors with particularly strong relationships to dementia as measured by the hazard ratio were age, a diagnosis of stroke, epilepsy or gait problems, use of SSRIs, receipt of social care, the presence of a third party at consultations and high numbers of missed GP appointments (Table 4). Risk reduced with increasing BMI and with higher systolic blood pressure. Excepting the correlation between age and age-squared, each factor in the final model had a variance inflation factor (VIF) of <2.0, generally considered to be low multicollinearity [45].

**Validation and calibration.** The 60–79 validation cohort consisted of 419,126 individuals. The mean age at Index (67.9 years; SD 6.4), sex distribution (48.5% male), median follow-up (2.67 years; IQR 0.99 to 5.0) and crude incident rate (6.31 per 1,000 person years) were all very similar to the development cohort. The validation and development cohorts were likewise closely alike with respect to all other predictive factors (Table 1).

When applied to the validation cohort, the reduced predictive model returned a Harrell's C of 0.781 (0.776 to 0.786), Royston D of 1.74 (1.70 to 1.78) and 1% (5%) precision of 36.7% (22.8%) (Table 3). The model had a calibration slope of 0.981 (0.956 to 1.006) indicating good calibration. A plot of observed risk against mean predicted risk within deciles of predicted risk demonstrated a reasonably linear relationship, with some slight over-estimation at the top end (Fig A2 in S1 File).

Re-fitting the reduced model using multi-level Cox regression produced predictive indices identical, or very similar, to those for the single-level model: Harrell's C = 0.781 (0.776 to 0.786); Royston D = 1.75 (1.71 to 1.79); top 1%(5%) precision = 36.7%(22.9%). C-statistics for individual GP practices in the validation cohort ranged from 0.63 to 0.99 with a median of 0.78, while 128 of the 158 practices (81%) had a C-statistic of 0.75 or above.

**Risk classification.** Table 5 summarises the discriminative performance of the reduced model in the validation sample, at different thresholds for defining "high risk" individuals. The table shows, for example, that 41.9% of individuals with a risk score of 20% or more went on to receive a dementia diagnosis compared to 5.5% of those with a lower risk score. Modelled rates taking into account censored cases were somewhat lower, at 29.9% versus 2.8% respectively. Rates modelled on the premise of annual risk assessments were very similar (Table A2 in S1 File).

**Table 4. Final predictive model for 60–79 cohort after backwards elimination.**

| | Coefficient | 95% CI | Hazard ratio | 95% CI |
|---|---|---|---|---|
| **Demographic and lifestyle factors** | | | | |
| Age | 0.268 | 0.249 to 0.287 | 1.308 | 1.283 to 1.333 |
| Age-squared | -0.006 | -0.006 to -0.005 | 0.994 | 0.994 to 0.995 |
| Practice IMD quintile | | | | |
| 1 Lowest deprivation | 0.0 | - | 1.000 | - |
| 2 | 0.001 | -0.110 to 0.111 | 1.001 | 0.896 to 1.118 |
| 3 | 0.012 | -0.101 to 0.124 | 1.012 | 0.904 to 1.132 |
| 4 | 0.071 | -0.046 to 0.188 | 1.073 | 0.955 to 1.207 |
| 5 highest deprivation | 0.221 | 0.099 to 0.343 | 1.247 | 1.104 to 1.409 |
| **Medical conditions** | | | | |
| History of depression | 0.121 | 0.063 to 0.179 | 1.128 | 1.065 to 1.196 |
| History of stroke | 0.486 | 0.432 to 0.539 | 1.625 | 1.541 to 1.715 |
| History of diabetes | 0.304 | 0.254 to 0.355 | 1.356 | 1.289 to 1.426 |
| History of epilepsy | 0.435 | 0.304 to 0.567 | 1.546 | 1.355 to 1.763 |
| History of gait problems | 0.414 | 0.316 to 0.513 | 1.513 | 1.372 to 1.670 |
| History of major head injury | 0.307 | 0.231 to 0.382 | 1.359 | 1.260 to 1.466 |
| **Prescribed medications (ever prescribed)** | | | | |
| Antidepressants | | | | |
| Selective serotonin reuptake inhibitor | 0.381 | 0.321 to 0.442 | 1.464 | 1.378 to 1.555 |
| Tricyclic antidepressant | -0.095 | -0.148 to -0.041 | 0.910 | 0.862 to 0.960 |
| Other antidepressants | 0.190 | 0.110 to 0.269 | 1.209 | 1.117 to 1.309 |
| Mood stabilisers | 0.227 | 0.143 to 0.312 | 1.255 | 1.154 to 1.366 |
| NSAIDs (excluding aspirin) | -0.158 | -0.203 to -0.114 | 0.853 | 0.816 to 0.892 |
| Anticholinergic burden over last year (square-root) | 0.136 | 0.107 to 0.165 | 1.146 | 1.113 to 1.180 |
| **Service interactions** | | | | |
| Ever received social services | 0.782 | 0.567 to 0.996 | 2.185 | 1.764 to 2.707 |
| Number of A&E visits in last year (square root) | 0.255 | 0.195 to 0.316 | 1.291 | 1.215 to 1.372 |
| Number of GP consultations involving third party in last year (square root) | 0.333 | 0.228 to 0.438 | 1.395 | 1.256 to 1.549 |
| Number of GP home visits in last year (square root) | 0.240 | 0.186 to 0.293 | 1.271 | 1.204 to 1.341 |
| Number of missed GP appointments in last year (square root) | 0.408 | 0.352 to 0.464 | 1.504 | 1.422 to 1.590 |
| **Biometric measures** | | | | |
| Most recent BMI (square root) | -0.486 | -0.537 to -0.435 | 0.615 | 0.585 to 0.647 |
| Most recent mean systolic blood pressure value/20[a] | -0.050 | -0.078 to -0.022 | 0.951 | 0.925 to 0.979 |

IMD, Index of Multiple Deprivation

[a]Rescaled by dividing by 20

**Comparison with other risk models.** A model based on the DRS factor-set but re-calibrated on our development cohort, had a C-statistic of 0.773 (0.768 to 0.779) for the validation cohort; only slightly below that obtained with the current study's factor set (Table A3 in S1 File). Royston's D was somewhat lower at 1.63 (1.59 to 1.67), as was the top 1% (5%) precision at 26.4% (19.3%).

A model using solely age and age-squared as risk factors had a Harrell's C of 0.735 (0.729 to 0.741) and Royston's D of 1.28 (1.24 to 1.32), both substantially below the DemRisk and DRS models (Table A3 in S1 File). The difference in performance was most clearly seen in the precision scores, where rates of future dementia amongst patients in the top 1% (5%) of risk scores were 36.7% (22.8%) for DemRisk versus 16.8% (11.6%) for the age-only model.

**Table 5. Risk classification table for the 60–79 validation cohort, at different thresholds for high risk of incident dementia.**

| Threshold for high risk | Cases classified as high risk, n(%) | Cases classified as low risk, n (%) | Sensitivity (%) | Specificity (%) | True Positives n (%) | False Negatives n (%) | Modelled TP[a] % | Modelled FN[a] % |
|---|---|---|---|---|---|---|---|---|
| 3% | 58,676 (45.2) | 71,281 (54.9) | 81.4 | 57.1 | 6,043 (10.3) | 1,382 (1.9) | 6.5 | 1.1 |
| 4% | 45,195 (34.8) | 84,762 (65.2) | 70.7 | 67.4 | 5,252 (11.6) | 2,173 (2.6) | 7.5 | 1.3 |
| 5% | 31,609 (24.3) | 98,348 (75.7) | 57.1 | 77.7 | 4,241 (13.4) | 3,184 (3.2) | 8.7 | 1.6 |
| 10% | 5,633 (4.3) | 124,324 (95.7) | 18.2 | 96.5 | 1,354 (24.0) | 6,071 (4.9) | 16.2 | 2.4 |
| 15% | 1,792 (1.4) | 128,165 (98.6) | 8.1 | 99.0 | 599 (33.4) | 6,826 (5.3) | 23.2 | 2.7 |
| 20% | 754 (0.6) | 129,203 (99.4) | 4.3 | 99.6 | 316 (41.9) | 7,109 (5.5) | 29.9 | 2.8 |
| 25% | 381 (0.3) | 129,576 (99.7) | 2.3 | 99.8 | 173 (45.4) | 7,252 (5.6) | 36.3 | 2.8 |

TP, True Positives (high risk cases who developed dementia); FN, False Negatives (low risk cases who developed dementia)

Table based on uncensored cases only; n = 129,957, rate of incident dementia within 5 years excluding censored cases = 5.7%

[a]TP and FN modelled to account for censored cases; modelled overall rate of incident dementia within 5 years accounting for censored cases = 3.2%

## Cohort aged 80–89 years

**Development cohort 80–89 years.** The development cohort included 175,131 individuals, 40.5% male, with a mean age at Index of 83.2 years (SD 2.92). The median length of follow-up was 1.81 years (inter-quartile range 0.70 to 3.83) and there were 15,994 incident cases of dementia over 397,710 person years giving a crude incidence rate of 40.2 per 1,000 person-years (Table A4 in S1 File). 18% were incident cases of Alzheimer's disease, 15% vascular dementia, and 67% mixed, unspecified or other. 40% of all incident cases were identified from the linked HES-APC dataset.

**Univariable associations between risk factors and incident dementia.** Associations between individual factors and incident dementia were generally weaker in the 80–89 age cohort (Table A5 in S1 File), though many of the strongest factors were the same: stroke, use of non-tricyclic antidepressants, receipt of social care, a third party at consultations and BMI.

**Multivariable analysis.** A Cox model based on the full set of factors had a Harrell's C-statistic of 0.646 (0.640 to 0.651), Royston D of 0.822 (0.788 to 0.856) and top 1% (5%) precision of 81.0% (69.4%), when applied to the development cohort (Table A6 in S1 File). The very high precision score was partly due to the much higher incidence of dementia in this age-group. After stepwise removal, a reduced model was obtained consisting of considerably fewer factors without greatly reducing performance in the development cohort. This model (Table 6) substantially overlapped with the factors in the younger cohort model. Other than age and age-squared, each factor in the model had a variance inflation factor (VIF) of <2.0.

**Validation and calibration.** The 80–89 validation cohort consisted of 118,717 individuals and was very similar to the development cohort on all measures (Table A4 in S1 File).

When applied to the validation cohort, the reduced predictive model returned a Harrell's C of 0.637 (0.630 to 0.643), Royston D of 0.737 (0.700 to 0.774) and 1% (5%) precision of 78.6% (71.0%) (Table A6 in S1 File). The model had a calibration slope of 0.973 (0.919 to 1.027) and the plot of observed risk against mean predicted risks within deciles of predicted risk demonstrated a reasonably linear relationship with slight over-estimation at the top end (Fig A2 in S1 File).

Results for the age 80–89 cohort reduced model using multi-level Cox regression were very close to those for the single-level model: Harrell's C = 0.637 (0.631 to 0.644); Royston D = 0.739 (0.707 to 0.771); top 1%(5%) precision = 78.2%(71.2%).

**Table 6. Final predictive model for 80–90 cohort after backwards elimination.**

| | Coefficient | 95% CI | Hazard ratio | 95% CI |
|---|---|---|---|---|
| **Demographic and lifestyle factors** | | | | |
| Female | 0.163 | 0.126 to 0.199 | 1.176 | 1.135 to 1.220 |
| Age—80 | 0.046 | 0.028 to 0.065 | 1.047 | 1.028 to 1.067 |
| (Age– 80)^2 | -0.004 | -0.006 to -0.002 | 0.996 | 0.994 to 0.998 |
| Practice IMD quintile | | | | |
| 1 Lowest deprivation | 0.000 | - | 1.000 | - |
| 2 | -0.020 | -0.112 to 0.073 | 0.981 | 0.894 to 1.075 |
| 3 | 0.016 | -0.077 to 0.109 | 1.016 | 0.926 to 1.115 |
| 4 | 0.039 | -0.054 to 0.133 | 1.040 | 0.947 to 1.142 |
| 5 highest deprivation | 0.126 | 0.027 to 0.224 | 1.134 | 1.028 to 1.252 |
| **Medical conditions** | | | | |
| History of stroke | 0.241 | 0.199 to 0.283 | 1.273 | 1.220 to 1.328 |
| History of diabetes | 0.151 | 0.104 to 0.198 | 1.163 | 1.110 to 1.219 |
| History of gait problems | 0.160 | 0.096 to 0.225 | 1.174 | 1.100 to 1.253 |
| History of major head injury | 0.230 | 0.167 to 0.293 | 1.258 | 1.181 to 1.340 |
| **Prescribed medications (ever prescribed)** | | | | |
| Antidepressants | | | | |
| Selective serotonin reuptake inhibitor | 0.320 | 0.275 to 0.366 | 1.378 | 1.316 to 1.442 |
| Tricyclic antidepressant | -0.079 | -0.120 to -0.038 | 0.924 | 0.887 to 0.962 |
| Other antidepressants | 0.177 | 0.107 to 0.246 | 1.193 | 1.113 to 1.279 |
| NSAIDs (excluding aspirin) | -0.145 | -0.185 to -0.105 | 0.865 | 0.831 to 0.900 |
| Anticholinergic burden over last year (square-root) | 0.047 | 0.023 to 0.071 | 1.048 | 1.023 to 1.073 |
| **Service interactions** | | | | |
| Ever received social services | 0.302 | 0.136 to 0.469 | 1.353 | 1.145 to 1.599 |
| Number of A&E visits in last year (square root) | 0.179 | 0.137 to 0.220 | 1.196 | 1.147 to 1.246 |
| Number of GP consultations involving third party in last year (square root) | 0.241 | 0.173 to 0.309 | 1.273 | 1.189 to 1.362 |
| Number of GP home visits in last year (square root) | 0.211 | 0.182 to 0.241 | 1.235 | 1.200 to 1.272 |
| Number of missed GP appointments in last year (square root) | 0.321 | 0.276 to 0.366 | 1.378 | 1.318 to 1.441 |
| **Biometric measures** | | | | |
| Most recent BMI (square root) | -0.408 | -0.447 to -0.368 | 0.665 | 0.639 to 0.692 |
| Most recent mean systolic blood pressure value/20[a] | -0.085 | -0.105 to -0.064 | 0.919 | 0.900 to 0.938 |

IMD, Index of Multiple Deprivation;

[a]Rescaled by dividing by 20

**Risk classification.** The classification table for the 80–89 cohort demonstrated high rates of both true positives and false negatives at all risk thresholds, reflecting the considerably higher rate of incident dementia in this older cohort. The rates modelled to account for censoring were reduced but still substantial (Table 7). Rates modelled on the premise of annual risk assessments were very similar (Table A7 in S1 File).

**Comparison with other models.** A model based on the DRS older cohort factor-set had performance indices in the validation cohort lower than those obtained with the current study's factor set: Harrell's C = 0.608 (0.602 to 0.614); Royston's D = 0.594 (0.558 to 0.629); top 1% (5%) precision = 67.0% (58.2%) (Table A8 in S1 File).

An age-only model had indices that were lower still: Harrell's C = 0.533 (0.527 to 0.539); Royston's D = 0.217 (0.185 to 0.249) (Table A8 in S1 File). Top 1% (5%) precision was only around half of the rate achieved by DemRisk, at 40.1% (36.7%) compared to 78.6% (71.0%).

**Table 7. Risk classification table for the 80–89 validation cohort, at different thresholds for high risk of incident dementia.**

| Threshold for high risk | Cases classified as high risk, n(%) | Cases classified as low risk, n (%) | Sensitivity (%) | Specificity (%) | True Positives n (%) | False Negatives n (%) | Modelled TP[a] % | Modelled FN[a] % |
|---|---|---|---|---|---|---|---|---|
| 10% | 30,209 (99.0) | 301 (1.0) | 99.4 | 1.2 | 11,007 (36.4) | 71 (23.6) | 20.5 | 9.2 |
| 15% | 22,558 (73.9) | 7,952 (26.1) | 83.0 | 31.2 | 9,192 (40.8) | 1,886 (23.7) | 23.0 | 12.9 |
| 20% | 11,317 (37.1) | 19,193 (62.9) | 51.8 | 71.3 | 5,742 (50.7) | 5,336 (27.8) | 28.3 | 15.4 |
| 25% | 5,518 (18.1) | 24,992 (81.9) | 29.6 | 88.5 | 3,278 (59.4) | 7,700 (31.2) | 34.1 | 17.0 |
| 30% | 2,922 (9.6) | 27,588 (90.4) | 17.4 | 94.9 | 1,924 (65.9) | 9,154 (33.2) | 39.7 | 18.0 |
| 40% | 923 (3.0) | 29,587 (97.0) | 6.2 | 98.8 | 683 (74.0) | 10,395 (35.1) | 50.5 | 19.2 |
| 50% | 319 (1.1) | 30,191 (98.9) | 2.3 | 99.7 | 252 (79.0) | 10,826 (35.9) | 60.8 | 19.8 |

TP, True Positives (high risk cases who developed dementia); FN, False Negatives (low risk cases who developed dementia)

Table based on uncensored cases only; n = 30,510, rate of incident dementia within 5 years excluding censored cases = 36.3%

[a]TP and FN modelled to account for censored cases; modelled overall rate of incident dementia within 5 years accounting for censored cases = 20.4%

## Discussion

We developed improved models to predict newly recorded dementia in primary care for patient groups aged 60–79 and 80–89, using information extracted from UK primary care EHRs. The model for individuals aged 60–79 demonstrated a good Harrell's C-statistic of 0.78 and good calibration in the validation cohort, while 81% of individual practices had a C-statistic of 0.75 or higher. The model for the older cohort had only a moderate C-statistic, but was twice as successful at identifying patients who would go on to get dementia compared to selection by age alone. Compared to the DRS factor set, both algorithms possessed greater ability to identify patients at high risk of incident dementia.

Three key elements of our study contribute to the improved predictive performance of our models. First, we investigated a larger pool of candidate risk factors, all with an evidence-based association with incident dementia. Second, our analysis cohorts consisted of data at a randomly selected age-point for each individual, resulting in an age distribution more representative of the patient population eligible for risk assessment in primary care. Third, we greatly increased the completeness of information on recorded dementia by linking in dementia diagnoses recorded in secondary care.

### Factors in the final models

**Model for 60–79 cohort.** Our final model for the younger cohort included a broad mix of factors. Age was the single strongest predictor of incident dementia, other strong predictors being a history of stroke or epilepsy, gait problems, anticholinergic burden, use of SSRIs, receipt of social services, third-party involvement in consultations and a high number of missed GP appointments. Factors associated with reduced risk included NSAIDS, higher BMI and higher blood pressure. These associations are all in line with existing evidence [46–49], though it is worth noting that obesity and hypertension measured in midlife predict future dementia and it is thought that weight loss and lower blood pressure in later life may potentially be symptoms of developing dementia [10]. Gender had very little association with incident dementia once controlled for other factors.

Diagnosed depression was a weak predictor, whereas use of SSRIs and anti-depressive drugs other than TCAs were strong predictors. TCAs themselves were negatively related: evidence around the association between TCA use and incident dementia is inconsistent [50] but

their anti-inflammatory properties may have protective benefits [50, 51]. We explored combining depression diagnosis and medications into a single variable, but fit indices substantially decreased. Inter-correlations between these factors were at most moderate and variance inflation factors low, hence multicollinearity does not seem an issue. Different types of anti-depressives may impact on dementia risk differently and independently of depressive symptoms themselves [50].

**Model for 80–89 cohort.** Our model for the older cohort mostly comprised a subset of the factors included in the younger cohort model, though associations with incident dementia were generally weaker. Gender was the only additional factor, implying an increased risk for older females; other studies have reported an increase of incident dementia in females from age 80 years [52].

## Strengths and limitations

Our development cohorts consisted of several hundred thousand individuals, including many thousands of incident cases of dementia, across more than 200 GP practices. Reasonably large validation cohorts provided accurate estimates of fit indices. We investigated a pool of 60 potential factors and our final models include up to 19 predictors. Large predictive models are at higher risk of overfitting. However, we think DemRisk unlikely to be overfitted. First, all of our factors were prespecified on the basis of possessing a good evidence base and only included in our final models if their direction of effect was in line with the literature; this limits the potential for overfitting through data-driven variable selection [53]. Second, the great majority of the included factors also feature in more than one previously published dementia prediction model. Third, our sample size analysis indicated that we had sufficient data to build models with up to 200 factors whilst keeping overfitting and other biases within acceptable limits.

We developed and validated our models using a randomly selected index date for each individual. This produced patient cohorts representative of the primary care population eligible for risk assessment across the time period, and thereby realistic estimates of the discriminative performance likely to be achieved in practice, particularly the rates of true positives and false negatives at various risk thresholds. We restricted our dataset to practices with linked secondary care data which may have affected the representativeness of the analysis sample; however, this was more than outweighed by the increased completeness of data on incident dementia compared to using diagnoses recorded in the EHR alone. While we believe our results generalisable to the wider English primary care population, they may not extend to other health systems.

When compared to the DRS recalibrated on our data, our models displayed better performance in most respects. For the 60–79 cohort the C-statistics differed only slightly, implying similar ability to rank-order pairs of patients; however, our model displayed a larger Royston D, indicating greater ability to separate higher- from lower-risk patients. Our model's Royston D was larger by 0.11, which compares to Royston's suggested criteria for an important difference of $> = 0.1$ [40]. This is reflected in the finding that our model had a top 1% precision of 37% compared to 27% for the DRS. A very similar pattern of results was found for the older cohort.

We note that some of our factors may be sensitive to prodromal or unrecorded dementia as well as to individuals purely at risk. These include depression and the measures of service interactions (e.g. third-party consultations, missed appointments) [31]. Considering that these patients had no prior codes for cognitive impairment or memory loss, they may still benefit from identification. Formal assessments may be required to distinguish cases of undetected dementia from strictly high risk individuals. Diagnosis usually occurs during the mid-stage of

dementia, typically two-to-four years after first onset [54], and our models are focused on picking up these individuals before they show clear symptoms. Early-stage dementia cannot be reliably identified through use of Read codes, as the key symptoms such as mild memory loss are common to many other conditions, including normal ageing. Identification of patients at risk of early-stage dementia may require a different form of approach.

To facilitate automated estimation of risk scores from the EHR alone we avoided predictive factors that require the collection of additional information from patients, are unreliably or poorly recorded, or only available from external sources. Even so, in everyday practice the records for 10% or more of patients may lack the information on BMI or blood pressure required to compute a risk score. In the absence of this information becoming available, we have found that a form of simple mean imputation may be applied to produce a reasonable approximation to a patient's actual risk score (See S1 File).

Calendar year did not feature in either the younger or older cohort model, suggesting little in the way of "calibration drift" or change in baseline risk over time [55]. Even so, in routine practice regular updating would be advisable.

### Relation to existing prediction models

The DemRisk models have been developed for deployment within UK primary care, and designed to be automated within the EHR system whilst minimising numbers of unrecorded or prodromal dementia cases amongst the identified patients. The only other prediction models currently validated in a UK population are the DRS and the two UK Biobank-based models [18, 19]. We have presented direct comparisons with the DRS, but neither Biobank model is amenable to full automation in primary care. The UKBDRS includes 11 factors of which 8 also appear in DemRisk and the DRS: age, sex, material deprivation, diabetes, stroke, depression, hypertension, and high cholesterol. The remaining factors of education, parental history of dementia and living alone are not recorded in the EHR except for specific patients, making full automation not possible. Furthermore, validation indices for the model—an AUC of 0.8 under internal validation in the UK Biobank and 0.77 under external validation in the Whitehall II study dataset—suggest predictive performance no better than that achieved by DemRisk or the DRS.

The second UK Biobank-based model, the UKB-DRP, is even less suited to deployment in primary care. This machine-learning based model consists of age, APOE4 gene, a card-pairs memorising game, leg fat percentage, number of medications, reaction time, peak expiratory flow, mother's age at death, long-standing illness, and mean corpuscular volume. The internal validation AUC was 0.85, but the memory test creates susceptibility to unrecorded or prodromal dementia, while most factors are not routinely available in the EHR and would require patients to undergo a battery of tests.

Numerous models exist that have been developed and validated outside of a UK context, but very few of these could be transported into UK primary care for current purposes, and those that could have at best only modest predictive ability. Thus a systematic review published up to April 2018, identified 38 unique models focused on an older (non-UK) general population [56], with risk factors numbering from 1 to 19 and AUCs/C-statistics from 0.62 to 0.91. However, 34 models incorporated subjective or objective measures of cognition such as the Mini Mental State Examination, while another two included factors not routinely available in UK primary care.

The two remaining, and potentially transportable, models overlap considerably with DemRisk and the DRS: the first was derived using the Framingham Heart Study dataset and consists of age, marital status, BMI, stroke, diabetes, ischemic heart attacks and cancer, but had a

low internal C-statistic of 0.72 [57]; the second utilised the Rotterdam Study dataset and consists of just age and sex [58]. The high internal AUC of this study, 0.79, seems likely to be sample-specific: both models were included in a Finnish large-scale external validation of 17 older general population models [59], and both had an external validity C-statistic of just 0.70. This study also highlighted that the only models to exceed a C-statistic of 0.75 were those containing cognition factors, whereas of 10 models with no such factors, none had an external C-statistic > 0.73. A literature search has not identified any newer transportable models with greater predictive ability. Better performing models are clearly required, and this may necessitate the discovery of novel predictive factors and/or larger models that can capture more of the differences between patients that contribute to their overall risk.

## Implications

The DemRisk models provide a useful contribution towards the development of a system for identifying patients at high risk of dementia from their EHR. When deployed in a practice record system, the models can be used to flag up patients for whom further clinical evaluation is advisable, including possibly referral to a memory assessment service depending upon the estimated level of risk and patient preferences. The choice of a risk threshold will depend upon the intended use, but as an example, using a threshold of 20% with the younger cohort, 3 in every 10 patients above this threshold are likely to develop dementia within 5 years, compared to 3 in every 100 below this threshold. Discriminative ability for the older cohort was lower, but our model nonetheless greatly outperformed selection by age alone and above a threshold risk score of 30% around 4 in every 10 patients aged 80 to 89 would go on to receive a dementia diagnosis. Details of how to compute risk scores from a patient's EHR using the algorithms are provided in S1 File.

Further research is needed to more firmly establish the external validity of the models. Although our sample sizes were large relative to the number of factors investigated and all the factors were evidence-based, our development and validation cohorts were both drawn from the same population of English GP practices using the Vision patient record system only. Validation work is strongly advised in relation to other countries in the UK and other record systems, such as the widely used EMIS and SystmOne. Although our models have been developed to maximise their performance in relation to the UK primary care system, research into their transportability to other countries and different healthcare systems would also be desirable. Other dementia risk prediction models, including the DRS, have demonstrated very mixed results in this respect [43, 60].

Finally, we note that many ethical concerns and practical challenges exist regarding the acceptability and implementation of dementia risk prediction within primary care services [61]. These complex issues should not be underestimated but are beyond the scope of the current paper.

## Conclusion

The DemRisk models can discriminate individuals at higher risk of dementia using only routinely collected data from their primary care record, and outperform an existing electronic-record based risk model. Discriminative ability was greatest for those aged 60 to 79 years, but the model for those aged 80 years plus may also be clinical useful. The models might best be used to identify patients for whom further clinical evaluation is desirable and to rule out those at very low risk. They could also play a role in helping to identify individuals at higher risk for invitation into trials of promising interventions.

## Supporting information

**S1 File. Additional information.**
(DOC)

**S2 File. Codelists.**
(DOCX)

**S1 Checklist. STROBE.**
(DOCX)

**S2 Checklist. TRIPOD.**
(DOCX)

## Acknowledgments

We wish to acknowledge Professor Kate Walters and Dr Sarah Hardoon, developers of the Dementia Risk Score, for sharing with us the Readcode lists they produced in relation to that project, and also other individual researchers who supplied us with their Readcode lists for specific predictive factors. We also want to thank John Langham and Charlotte Wu for their insightful comments and suggestions as advisors to the study.

## Author Contributions

**Conceptualization:** David Reeves, Daniel Stamate, Elizabeth Ford, Darren M. Ashcroft, Evangelos Kontopantelis, Harm Van Marwijk.

**Data curation:** Catharine Morgan, Daniel Stamate.

**Formal analysis:** David Reeves, Catharine Morgan, Daniel Stamate.

**Funding acquisition:** David Reeves, Daniel Stamate, Elizabeth Ford, Darren M. Ashcroft, Evangelos Kontopantelis, Harm Van Marwijk.

**Methodology:** David Reeves, Catharine Morgan, Daniel Stamate, Elizabeth Ford, Darren M. Ashcroft, Evangelos Kontopantelis, Brian McMillan.

**Project administration:** David Reeves, Daniel Stamate.

**Resources:** David Reeves, Catharine Morgan, Daniel Stamate, Brian McMillan.

**Software:** Catharine Morgan, Daniel Stamate.

**Supervision:** David Reeves, Daniel Stamate.

**Writing – original draft:** David Reeves, Catharine Morgan.

**Writing – review & editing:** David Reeves, Catharine Morgan, Daniel Stamate, Elizabeth Ford, Darren M. Ashcroft, Evangelos Kontopantelis, Harm Van Marwijk, Brian McMillan.

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
