## [Decision Letter · Decision Letter 0]

25 Jun 2024

PONE-D-24-19444Identifying individuals at high risk for dementia in primary care: development and validation of the DemRisk risk prediction model using routinely collected patient dataPLOS ONE

Dear Dr. Reeves,

Thank you for submitting your manuscript to PLOS ONE. After careful consideration, we feel that it has merit but does not fully meet PLOS ONE’s publication criteria as it currently stands. Therefore, we invite you to submit a revised version of the manuscript that addresses the points raised during the review process.

We look forward to receiving your revised manuscript.

Kind regards,

Aamna AlShehhi, PhD

Academic Editor

PLOS ONE

Journal Requirements:

Reviewers' comments:

Reviewer's Responses to Questions

**Comments to the Author**

1. Is the manuscript technically sound, and do the data support the conclusions?

Reviewer #1: Partly

Reviewer #2: Partly

2. Has the statistical analysis been performed appropriately and rigorously? 

Reviewer #1: No

Reviewer #2: Yes

3. Have the authors made all data underlying the findings in their manuscript fully available?

Reviewer #1: No

Reviewer #2: Yes

4. Is the manuscript presented in an intelligible fashion and written in standard English?

Reviewer #1: Yes

Reviewer #2: Yes

5. Review Comments to the Author

**Reviewer #1:** Line 215 - it would be beneficial if the time period for polypharmacy and pulse pressure are given int he text (not just tables).

Line 225 - there are grave statistical consequences of using a missing data category instead of imputing. Missing data indicator is no longer encouraged when analysing data It is possible to see that the amount of missing data for the health indicators is not high, and most older people will have these data collected regularly due to them having at least one QoF condition. The Walters, Hardoon paper showed that it was possible to impute these data adequately. For smoking status, those that are missing are most likely to be non-smokers https://bmjopen.bmj.com/content/bmjopen/4/4/e004958.full.pdf

Line 253 - the outcome is time to dementia diagnosis

In the participants section, you say you exclude those with less than one year follow-up, but the lower quartile follow-up is 0.96, so assume these and the 25% with shorter follow-up got dementia in the first year of follow-up?

Table 1 - sight impairment must be underrecorded as many people of that age would wear glasses for some degree of sight impairment, but many will not consult their GP for conditions that can be helped by opticians.

Table 1 - define GP consultations involving a 3rd party (it only becomes when reading univariable associations text).

Table 1 - define anticholinergic burden.

Table 2 - no need to put ns in the table. It is better to allow readers to make their own minds up from the estimate and 95% CI.

Line 364 - "The two cohorts..." this sentence sounds as if it has not been checked. If it has not been checked then remove (or check it). If it has been checked then modify the sentence to make it clear.

Line 398 (and beginning of the paragraph) - I think total number of participants or person years is wrong as they read as the same.

Line 466 - Higher blood pressure in midlife is associated with higher dementia risk. Blood pressure decreases in later life (usually) so lower blood pressure measured in later life as here is a risk factor for dementia. See Lancet Commission for Dementia, 2020.

**Reviewer #2:** Comments to Authors:

The aims of the current study were to develop risk models with improved ability to identify patients at higher risk of future dementia from their UK primary care record, by investigating a wider set of potential predictive factors, constructing development and validation cohorts more representative of the target population, and by increasing the completeness of information on dementia diagnosis using linked secondary care data. This is a robust study, and I read with great interest. However, I have a few concerns.

- Major points

1. Generally, patients who visit GPs are at higher risk of future dementia or may have great interest in preventing dementia. Namely, the authors’ study may have selection bias. They may be moderate or severe dementia cases. Mild cases may not be recorded in the GPs EHRs.

2. According to IQR of follow-ups, about a quarter of patients for development cohort 60-79 years (0.96, 5) and validation cohort 60-79 years (0.99, 5) were lost to follow-up before a year (Table 1). Characteristics of early censored participants could be different from participants who followed up 5 years. Some of them might develop dementia when followed up longer.

3. Prescribed medications could have overlapped with some medical conditions (ever recorded). When both sets of variables are entered in the same model, multicollinearity is expected. Usually, working operational definitions are developed by combining records of medication and medical conditions.

4. Although CPRD has unreliable information such as education for older patients, education is a major factor in elucidating the relationship with dementia. Is there any proxy measure that can be used instead of education?

5. IMD represents geographic communities. The authors may consider additionally applying multilevel Cox proportional hazards regression models (line 254, 259).

6. With 60 evidence-based risk factors, the DemRisk models might be overfitted. This may lead to lack of generalizability. Models should be parsimonious. I suggest Chapter 7, Algorithms to Live By: The Computer Science of Human Decisions, written by Brian Christian and Tom Griffiths.”

- Minor points

1. The authors described their data source in detail so that readers can easily understand CPRD GOLD. Although this is a novel study, the authors may summarize a bit more (line 134-141, 142-149, 150-157).

2. Compared to the long list of study strengths and their implications, the discussion provided in the study is somewhat short. The explanation may not provide strong evidence for the authors’ large prediction models. Would you please add more evidence by reviewing other previous studies?

6. PLOS authors have the option to publish the peer review history of their article (what does this mean?). If published, this will include your full peer review and any attached files.

Reviewer #1: No

Reviewer #2: No

---

## [Author Response · Author response to Decision Letter 0]

23 Aug 2024

RE: Identifying individuals at high risk for dementia in primary care: development and validation of the DemRisk risk prediction model using routinely collected patient data

Please find our point-by-point response to the Editor and reviewer’s comments below

Journal Requirements:

https://journals.plos.org/plosone/s/file?id=wjVg/PLOSOne_formatting_sample_main_body.pdf [journals.plos.org] and 

https://journals.plos.org/plosone/s/file?id=ba62/PLOSOne_formatting_sample_title_authors_affiliations.pdf [journals.plos.org]

We have done our best to ensure that our manuscript meets the journal’s style requirements, including file naming.

2. Please note that PLOS ONE has specific guidelines on code sharing for submissions in which author-generated code underpins the findings in the manuscript. In these cases, we expect all author-generated code to be made available without restrictions upon publication of the work. Please review our guidelines at https://journals.plos.org/plosone/s/materials-and-software-sharing#loc-sharing-code [journals.plos.org] and ensure that your code is shared in a way that follows best practice and facilitates reproducibility and reuse.

We have posted all of the Read code-lists required to construct the variables used in the study from the CPRD dataset onto the GitHub public depository and provided the URL for this in our Data Availability statement, as copied below: 

“The data used in this study was obtained via the Clinical Practice Research Datalink (CPRD). CPRD provides a research service which provides representative, longitudinal real-time anonymized patient electronic health records data from primary care and other health services across the UK. The licensing agreement between University of Manchester and CPRD, and the data governance of CPRD prevent the sharing or distribution of patient data to other individuals. Hence any requests for access to data from the study should be addressed to cprdenquiries@mhra.gov.uk. All researchers requiring access will require approval of their proposals from CPRD before data release.

The Read code-lists used to generate all the variables used in the study from the CPRD GOLD dataset have been made publicly available at: https://github.com/C5thyM/demrisk.”

 We have added the ethics statement to the Methods section: 

“Access to CPRD GOLD was obtained under licence from the UK Medicines and Healthcare products Regulatory Agency. CPRD GOLD consists of routinely collected data that has been pseudonymised for the purposes of research, for which informed consent was not required. The study was approved by the independent scientific advisory committee for Clinical Practice Research Datalink research (protocol No 18_163R).”

Reviewers' comments:

Reviewer's Responses to Questions

Comments to the Author

5. Review Comments to the Author

Reviewer #1: Line 215 - it would be beneficial if the time period for polypharmacy and pulse pressure are given in the text (not just tables).

We have revised the text to include this information, which – due to other changes – now appears on lines 219 and 226.

Line 225 - there are grave statistical consequences of using a missing data category instead of imputing. Missing data indicator is no longer encouraged when analysing data It is possible to see that the amount of missing data for the health indicators is not high, and most older people will have these data collected regularly due to them having at least one QoF condition. The Walters, Hardoon paper showed that it was possible to impute these data adequately. For smoking status, those that are missing are most likely to be non-smokers https://bmjopen.bmj.com/content/bmjopen/4/4/e004958.full.pdf [bmjopen.bmj.com]

The debate around the use of the Missing Value Method (MVM) is not as clear cut as the reviewer suggests (see below). Nonetheless, we re-visited our decision and ran an analysis using multiple imputation (MI) (fully conditional specification with 10 imputed datasets) for comparison purposes. We discovered that the two methods produced identical C-statistics (0.781 for the 60-79 cohort) and Royston Ds (1.74), plus risk factor coefficients that in almost every case were near-identical. This indicated to us that the missingness was non-informative (see below). On this basis we decided to change the manuscript to report the results obtained using MI, since the models are slightly simpler without the missing value indicators and the method has greater “face validity”. 

However, this means that were our models to be deployed in practice, a separate method would be required to derive risk scores for cases with missing values (for various reasons MI itself cannot be deployed, not least because the Fully Conditional Specification method requires knowledge of the future outcome). To address this we now include a passage in the S1 File demonstrating a simple mean imputation method that produces reasonably accurate risk scores for more than 99% of these cases.

For context: the main criticism of MVM is that it can produce biased estimates of the co-effs for some risk factors. However, other authors point out that while MI will produce more accurate coefficients when the assumption that data is missing at random is correct, MI itself will be biased if missingness is informative. Recently, Ness et al (https://doi.org/10.1145/3580305.3599911) demonstrated both theoretically and empirically that in the presence of informative missingness MVM has greater predictive performance compared to imputation methods, and when missingness is not informative the inclusion of missing value indicators does not negatively affect performance. They recommend that when predictive performance is prioritised over precise imputations “MVM plus mean imputation is a strong method while being much more efficient than more complicated imputation schemes.” We mention this only to demonstrate that we had a good rationale for our original analysis decision in this respect.

Having now adopted the MI method, missing data on smoking status has been subjected to multiple imputation, hence there is no necessity to collapse this category with another. 

Line 253 - the outcome is time to dementia diagnosis

We have revised the first sentence of the paragraph to read “We applied Cox proportional hazards regression models to build our predictive models, using time to dementia diagnosis as the outcome.” (new Line 259)

In the participants section, you say you exclude those with less than one year follow-up, but the lower quartile follow-up is 0.96, so assume these and the 25% with shorter follow-up got dementia in the first year of follow-up?

We now realise that our text was in error. We did not exclude patients with less than one year of follow-up; rather, we excluded patients with less than one full year of EHR data prior to their date of study entry. The reviewer is correct in saying that 25% had less than one year of follow-up, but not all of these got dementia in the first year of follow-up: some died or moved away from the practice, or the practice may have ceased contributing to CPRD. However, all patients had a minimum of one full year of data prior to the date of study entry (ie the start of follow-up). We have revised the text to read “We excluded patients with a code for dementia recorded prior to study entry, with less than one year of continuous registration in the practice prior to study entry, or with less than one full year of consultation data prior to study entry.” (new line 170)

Table 1 - sight impairment must be under-recorded as many people of that age would wear glasses for some degree of sight impairment, but many will not consult their GP for conditions that can be helped by opticians.

Many conditions are under-recorded in the EHR. This is an acknowledged limitation of the data and may be one reason why some factors do not appear in the algorithms. However, we do expect most of the more severe cases of sight impairment to be recorded, where for example opticians have been unable to help or have advised a person to consult their GP, as they are advised to do in the UK.

Table 1 - define GP consultations involving a 3rd party (it only becomes when reading univariable associations text).

We have revised the text in the table to read “Number of GP consultations involving third party (e.g. relative/friend) in last year”.

Table 1 - define anticholinergic burden.

We have added a definition and related reference to the main text at line 213, in the passage about predictive factors: “We also included measures of polypharmacy (number of different prescribed medicines over previous 12 months) and of anticholinergic burden (by which each drug is assigned a score of 1, 2 or 3 depending upon degree of anticholinergic effect and the scores totalled.(33)”

Table 2 - no need to put ns in the table. It is better to allow readers to make their own minds up from the estimate and 95% CI.

We have removed the ns from table 2. We have also removed them from the corresponding table (A5) in the S1 File for the older patient cohort

Line 364 - "The two cohorts..." this sentence sounds as if it has not been checked. If it has not been checked then remove (or check it). If it has been checked then modify the sentence to make it clear.

For clarity we have revised the sentence to read “The validation and development cohorts….” (new line 381)

Line 398 (and beginning of the paragraph) - I think total number of participants or person years is wrong as they read as the same.

We thank the reviewer for spotting this mistake. We have corrected the number of participants to be 175,131

Line 466 - Higher blood pressure in midlife is associated with higher dementia risk. Blood pressure decreases in later life (usually) so lower blood pressure measured in later life as here is a risk factor for dementia. See Lancet Commission for Dementia, 2020.

We have revised the text to include this point: “These associations are all in line with existing evidence,(46-49) though it is worth noting that obesity and hypertension measured in midlife predict future dementia and it is thought that weight loss and lower blood pressure in later life may potentially be symptoms of developing dementia.(10)” (new line 501)

Reviewer #2: Comments to Authors:

The aims of the current study were to develop risk models with improved ability to identify patients at higher risk of future dementia from their UK primary care record, by investigating a wider set of potential predictive factors, constructing development and validation cohorts more representative of the target population, and by increasing the completeness of information on dementia diagnosis using linked secondary care data. This is a robust study, and I read with great interest. However, I have a few concerns.

- Major points

1. Generally, patients who visit GPs are at higher risk of future dementia or may have great interest in preventing dementia. Namely, the authors’ study may have selection bias. They may be moderate or severe dementia cases. Mild cases may not be recorded in the GPs EHRs.

The GOLD dataset includes data on all patients registered with participating GPs - together with data on all their consultations at the GP practice for any reason - and almost all adults in the UK are registered with a GP practice. Our cohorts included all registered patients who at baseline had no symptoms of dementia or prodromal dementia (eg memory loss codes); there was no selection on the basis of the future outcome. Thus our sample includes all patients who went on to develop mild dementia as well as more severe forms. 

However, it is very likely that mild cases were less likely to be classed as positive cases for the purpose of model building. There is no easy or reliable way to identify mild dementia through use of Read codes: many of the related symptoms such as mild memory loss are shared by many other conditions, even just normal ageing. For this reason positive cases were defined (as they have been in other EHR-based studies) as those with a future diagnosis that specifically stated dementia, or prescription of a dementia-specific drug. We have added some relevant text to the strengths and limitations section, at line 552:

“Diagnosis usually occurs during the mid-stage of dementia, typically two-to-four years after first onset,(54) and our models are focused on picking up these individuals before they show clear symptoms. Early-stage dementia cannot be reliably identified through use of Read codes, as the key symptoms such as mild memory loss are common to many other conditions, including normal ageing. Identification of patients at risk of early-stage dementia may require a different form of approach.”

2. According to IQR of follow-ups, about a quarter of patients for development cohort 60-79 years (0.96, 5) and validation cohort 60-79 years (0.99, 5) were lost to follow-up before a year (Table 1). Characteristics of early censored participants could be different from participants who followed up 5 years. Some of them might develop dementia when followed up longer.

We agree. Early censored participants are more likely to be older and more ill and to have died in the first year of follow-up. Many patients who were censored due to leaving their practice, say, could have developed dementia at a later date. It is important to note that the Cox analysis does not assume that censored patients never developed dementia: these patients are assumed to come from the same population as patients with similar characteristics who remained with their practice. This is one of the main advantages of the Cox model, in that it models length of survival (in this case, time to dementia diagnosis) taking account of such covariates. 

3. Prescribed medications could have overlapped with some medical conditions (ever recorded). When both sets of variables are entered in the same model, multicollinearity is expected. Usually, working operational definitions are developed by combining records of medication and medical conditions.

We coded medications as different factors from medical conditions on the recommendation of Prof Darren Ashcroft, Professor of Pharmacoepidemiology and a member of the research team with long experience of analysing CPRD. Many medications are used for a variety of conditions and not restricted to a single one. In addition, many have biological effects that are distinct from (and can be opposite to) the effects of the medical condition itself. Our paper includes a good example in the case of depression and related medications. Diagnosed depression was a weak predictor of future dementia, whereas use of SSRIs and anti-depressive drugs other than TCAs were strong predictors. TCAs themselves were negatively related, which may be related to their anti-inflammatory properties. Notably, when we explored combining depression diagnosis and medications into a single variable fit indices substantially decreased, supporting their use as separate factors. 

Multicollinearity was an issue that we took into account during the stepwise reduction procedure in deciding between factors of similar strength for removal. We have revised the text at line 277 to read “The reduction was curated in that decisions about factors to be dropped or retained at each step were based not only on statistical considerations (p-values, hazard ratios and multicollinearity), but also on clinical and practical considerations

---

## [Decision Letter · Decision Letter 1]

6 Sep 2024

Identifying individuals at high risk for dementia in primary care: development and validation of the DemRisk risk prediction model using routinely collected patient data

PONE-D-24-19444R1

Dear Dr. David Reeves,

We’re pleased to inform you that your manuscript has been judged scientifically suitable for publication and will be formally accepted for publication once it meets all outstanding technical requirements.

Kind regards,

Aamna AlShehhi, PhD

Academic Editor

PLOS ONE

Reviewers' comments:

Reviewer's Responses to Questions

**Comments to the Author**

1. If the authors have adequately addressed your comments raised in a previous round of review and you feel that this manuscript is now acceptable for publication, you may indicate that here to bypass the “Comments to the Author” section, enter your conflict of interest statement in the “Confidential to Editor” section, and submit your "Accept" recommendation.

Reviewer #1: All comments have been addressed

Reviewer #2: All comments have been addressed

2. Is the manuscript technically sound, and do the data support the conclusions?

Reviewer #1: Yes

Reviewer #2: Yes

3. Has the statistical analysis been performed appropriately and rigorously? 

Reviewer #1: Yes

Reviewer #2: Yes

4. Have the authors made all data underlying the findings in their manuscript fully available?

Reviewer #1: No

Reviewer #2: Yes

5. Is the manuscript presented in an intelligible fashion and written in standard English?

Reviewer #1: Yes

Reviewer #2: Yes

6. Review Comments to the Author

Reviewer #1: (No Response)

Reviewer #2: To authors:

The manuscript was extensively revised based on comments. I am grateful to the authors for adding qualities to this excellent paper.

7. PLOS authors have the option to publish the peer review history of their article (what does this mean?). If published, this will include your full peer review and any attached files.

Reviewer #1: No

Reviewer #2: No

---

## [Editor Report · Acceptance letter]

24 Sep 2024

PONE-D-24-19444R1 

PLOS ONE

Dear Dr. Reeves, 

I'm pleased to inform you that your manuscript has been deemed suitable for publication in PLOS ONE. Congratulations! Your manuscript is now being handed over to our production team.

Kind regards, 

on behalf of

Dr Aamna AlShehhi 

Academic Editor

PLOS ONE